# Uncovering the Gradient Geometry of Long CoT: A Spectral-guided Approach to Reasoning Distillation

**Sinan Fan** [1 2] **Xiaofeng Sun** [1 2] **Chen Shen**[† 3] **Chenxi Huang** [3] **Shaotian Yan** [3] **Bing Wang** [3] **Kaiyuan Liu** [1 3]
**Xiaosong Yuan** [3] **Liang Xie** [4 3] **Wenxiao Wang** [1 3] **Jun Zhang** [5] **Hongyang Chen**[‡ 2] **Jieping Ye** [3]

## Abstract

Large reasoning models (LRMs) achieve remarkable reasoning performance by generating long chains-of-thought (CoT). However, standard supervised fine-tuning (SFT) treats all tokens uniformly, indiscriminately minimizing loss across both essential reasoning steps and those that are noisy, redundant, or instance-specific. This often leads student models to memorize superficial patterns rather than acquire generalizable reasoning capabilities. To better understand this limitation, we introduce *Loss Subspace Attribution*, a gradient decomposition analysis approach that uncovers a striking geometric structure: Gradients corresponding to effective reasoning predominantly lie within a low-rank consensus subspace, while conflicting or unstructured signals dominate the residual subspace. Guided by this insight, we propose ***Spectral-guided Learning***, a step-level distillation strategy that uses spectral strength to identify reasoning steps aligned with the consensus subspace and prioritizes their contribution to parameter updates, while suppressing gradients from the residual subspace. Experiments across various LRMs and diverse complex reasoning tasks consistently demonstrate that focusing optimization on the consensus subspace yields more robust and generalizable student models.

## 1. Introduction

Recent advancements have demonstrated that knowledge distillation from strong large reasoning models (LRMs) can

[1]Zhejiang University [2]Zhejiang Lab [3]Tongyi Lab, Alibaba Group [4]College of Computer Science and Technology, Zhejiang University of Technology [5]Department of Mathematics, University of Michigan. Correspondence to: Hongyang Chen <hongyang@zhejianglab.org>, Chen Shen <jason.sc@alibaba-inc.com>.

*Proceedings of the $43^{rd}$ International Conference on Machine Learning*, Seoul, South Korea. PMLR 306, 2026. Copyright 2026 by the author(s).

*Figure 1.* **An illustrative CoT with different reasoning steps**, including logic reasoning steps (blue blocks) and redundant steps (red blocks).

substantially empower smaller models with complex reasoning capabilities. Notably, DeepSeek (DeepSeek-AI et al., 2025) showed that directly fine-tuning dense student models on reasoning trajectories generated by superior teachers, such as DeepSeek-R1, yields remarkable performance gains. Driven by the simplicity of this supervised fine-tuning (SFT) paradigm and the deployment efficiency of smaller student models, there has been a resurgence of interest in distillation-based reasoning enhancement. In this strategy, the extended chains-of-thought (CoT) (Wei et al., 2022) produced by teacher models serves as the primary supervisory signal. However, current CoT distillation paradigms almost universally adopt a behavioral cloning approach: treating the complete CoT generated by teacher models as the gold standard and minimizing next-token prediction loss across all tokens.

However, as reasoning chains become longer, the limitations of treating all tokens equally become increasingly evident. Empirical studies (Sui et al., 2025; Wang et al., 2025b), as illustrated in Figure 1, have shown that only a small fraction of steps in a long CoT carry essential reasoning logic, while the remaining portions often consist of redundant elabo-

rations, filler phrases, or even erroneous attempts that get corrected later. Training on such verbose trajectories forces models to overfit to surface patterns that do not transfer across samples, thereby reducing the efficiency of reasoning distillation (Lin et al., 2024; Wu et al., 2025). Previous work has attempted to filter tokens based on statistical heuristics such as perplexity or entropy (Cui et al., 2025; Li et al., 2025; Zhang et al., 2025; Wang et al., 2025a). However, these metrics primarily reflect surface-level prediction uncertainty, assuming that lower uncertainty equates to higher data quality and thereby overlooking the essential properties required for generalizable learning.

To capture these essential properties, we approach the problem from an optimization perspective to analyze how the model internalizes reasoning patterns. We pioneer a gradient spectral analysis perspective in this domain and propose **Loss Subspace Attribution**, a gradient decomposition approach that bridges the gap between optimization geometry and reasoning semantics. Instead of treating gradients as opaque vectors, our framework leverages spectral decomposition to disentangle the gradient space into orthogonal subspaces and projects these geometric components back into the textual domain to identify which reasoning segments drive the model's generalization.

After conducting comprehensive analyses across multiple reasoning tasks and models, we discovered that: ***The gradient energy of long CoTs is highly concentrated in a low-rank consensus subspace, supported by reasoning steps with high spectral strength, rather than being uniformly distributed across all steps. Furthermore, gradients produced by high spectral strength steps maintain structured alignment across samples, whereas those from low spectral strength steps appear largely unstructured and weakly aligned.***

These findings, revealed by our *Loss Subspace Attribution*, indicate a stable separation between a low-rank consensus component and a residual component in gradient space. Our further analysis (Section 2.3) reveals that the residual gradients, i.e., the components orthogonal to the consensus directions, from low-spectral-strength steps exhibit significantly lower pairwise alignment across instances, often producing conflicting update directions. Such gradient interference is a known bottleneck for optimization efficiency (Yu et al., 2020; Dandi et al., 2022; Sankararaman et al., 2020), and can be mitigated by suppressing the influence of these residual components during training.

Based on the above observations, we propose **Spectral-guided Learning**, a selective learning approach that relies on spectral decomposition of gradient subspaces to guide optimization. Our method employs a two-stage process: spectral analysis and selective training. In the first stage, we perform gradient spectral decomposition on

CoT data using the student's initial weights, computing step-level spectral strength to identify steps that drive the consensus gradient subspace. During formal SFT training, we prioritize parameter updates from segments with high spectral strength, while suppressing residual gradients that offer limited transferable benefit.

To validate our analysis and evaluate Spectral-guided Learning, we conduct comprehensive experiments across multiple models and long CoT reasoning tasks. Experimental results show that, compared to standard SFT and heuristic filtering methods such as PPL and entropy (Cui et al., 2025; Li et al., 2025), our method consistently improves reasoning performance, achieving an average accuracy gain of 5.5% across diverse complex reasoning datasets. These results align with our observations and demonstrate the effectiveness of our method.

Overall, the main contributions of this paper are as follows:

- We are the first to investigate the training dynamics of long CoT reasoning through the lens of gradient spectral analysis. We propose the **Loss Subspace Attribution**, which analyzes training signals in long CoT reasoning via spectral analysis of gradients.

- We introduce spectral strength as a step-level metric to quantify the contribution of each reasoning step to the dominant gradient subspace. Our analysis reveals that gradient energy is highly concentrated in a low-rank consensus subspace supported by high spectral strength steps, while residual gradients from low spectral strength steps are largely instance-specific and serve as the dominant source of gradient conflicts.

- Based on the above analysis, we propose **Spectral-guided Learning**, a two-stage selective learning approach that performs gradient spectral decomposition offline and prioritizes parameter updates from high spectral strength segments during SFT, effectively suppressing residual gradients and mitigating cross-instance gradient conflicts.

- We conduct comprehensive experiments across multiple models and long CoT reasoning datasets, demonstrating that our method consistently improves reasoning performance over standard SFT and heuristic filtering methods, such as PPL.

## 2. Loss Subspace Attribution Framework

To understand which parts of the long CoT contribute to generalizable reasoning ability, we approach the problem from an optimization perspective, bridging the gap between optimization geometry and reasoning semantics to analyze how reasoning patterns are internalized through gradient updates.

## 2.1. Do Long CoT Gradients Exhibit Intrinsic Structure?

Raw gradients are high-dimensional and opaque. To disentangle the meaningful semantic signals from this complexity, we first investigate their underlying geometric structure.

**Setting.** To investigate the intrinsic structure of reasoning signals, we design a comparative experiment on the $1,000$ sampled instances using two distinct supervision views: the ***Reasoning CoT view*** (supervising the complete reasoning process) and the ***Non-Reasoning view*** (supervising only the final conclusion answer). For both views, we apply Singular Value Decomposition (SVD) to the gradient matrices to analyze their energy distributions, which quantifies the effective rank and reveals whether the optimization information is concentrated in a few core patterns or spread uniformly across dimensions. To formalize this analysis, we first define the loss-gradient embedding $g_t$ as the gradient of the loss at position $t$ with respect to the representation $\boldsymbol{h}_t$:

$$g_t = \nabla_{\boldsymbol{h}_t} \ell_t \in \mathbb{R}^d, \qquad (1)$$

where $\ell_t$ is the cross-entropy loss at position $t$ and $d$ is the hidden dimension. We stack the gradient embeddings of all $T$ tokens row-wise to form the gradient matrix:

$$\boldsymbol{G} = [g_1, g_2, \ldots, g_T]^\top \in \mathbb{R}^{T \times d}. \qquad (2)$$

We then perform SVD on the gradient matrix $\boldsymbol{G}$:

$$\boldsymbol{G} = \boldsymbol{U}\boldsymbol{\Sigma}\boldsymbol{V}^\top, \qquad (3)$$

where $\boldsymbol{U} \in \mathbb{R}^{T \times r}$ and $\boldsymbol{V} \in \mathbb{R}^{d \times r}$ are the left and right singular vector matrices, and $\boldsymbol{\Sigma} = \text{diag}(\sigma_1, \sigma_2, \ldots, \sigma_r)$ contains the singular values with $\sigma_1 \geq \sigma_2 \geq \cdots \geq \sigma_r > 0$. To quantify the energy distribution across singular directions, we define the cumulative energy function:

$$E(k) = \frac{\sum_{i=1}^{k} \sigma_i^2}{\sum_{j=1}^{r} \sigma_j^2}, \qquad (4)$$

which measures the fraction of total gradient energy captured by the first $k$ singular directions. If $E(k)$ approaches 1 for small $k$, the gradient matrix exhibits strong low-rank structure.

**Observation.** As illustrated in Figure 2, the cumulative energy curves reveal a clear difference between the two supervision views. The Reasoning CoT gradients exhibit a sharp "inverted-L" distribution, where a small fraction of principal directions captures most of the total energy. In contrast, the non-reasoning gradients rise more gradually, indicating that their energy is spread across a larger number of directions. This suggests that while long CoT sequences contain many tokens, their optimization signals are sparse and exhibit strong low-rank structure. Based on this observation, we decompose the gradient space into two orthogonal parts:

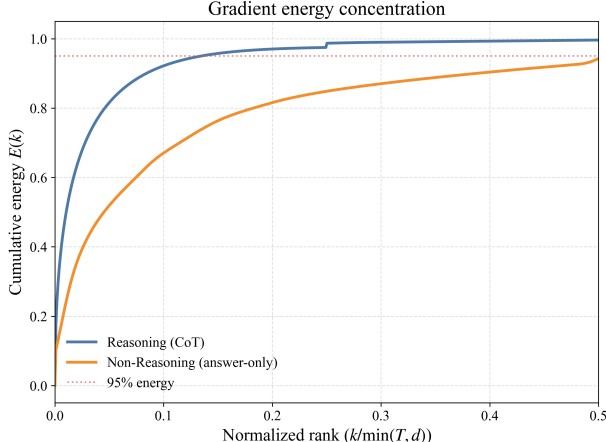

Gradient energy concentration

*Figure 2.* **An illustration of gradient spectral properties in reasoning vs. non-reasoning contexts.** Plotting cumulative energy ($y$-axis) against normalized rank ($x$-axis), we observe that the Reasoning (CoT) gradients (blue) are highly concentrated, crossing the 95% energy threshold significantly earlier than the Non-Reasoning baseline (orange).

- **Consensus Subspace.** Spanned by the first $k^*$ right singular vectors $\boldsymbol{V}_{1:k^*}$ that carry the principal energy. The projection operator is $\boldsymbol{P}_\| = \boldsymbol{V}_{1:k^*}\boldsymbol{V}_{1:k^*}^\top$, and we define the consensus gradient as:

$$g_t^\| = \boldsymbol{P}_\| g_t. \qquad (5)$$

  This component represents shared reasoning signals in the data.

- **Residual Subspace.** The orthogonal complement of the consensus subspace. The corresponding projection operator is $\boldsymbol{P}_\perp = \boldsymbol{I} - \boldsymbol{P}_\|$. We define the projection of gradients onto this space as the residual gradient:

$$g_t^\perp = \boldsymbol{P}_\perp g_t. \qquad (6)$$

## 2.2. How to Identify Reasoning Steps that Drive Consensus?

The previous subsection establishes that effective training signals are concentrated in a low-rank Consensus Subspace. While this subspace characterizes the global optimization direction, leveraging it for efficient learning requires identifying which specific reasoning steps contribute to it. Therefore, in this subsection, we propose a metric to map this global geometric structure back to local reasoning steps.

**Setting.** To quantify this contribution, we consider the projection energy of a token's gradient onto the consensus subspace, i.e., $\|g_t^\|\|^2$. By the SVD decomposition $\boldsymbol{G} = \boldsymbol{U}\boldsymbol{\Sigma}\boldsymbol{V}^\top$, this energy is intrinsically encoded in the row norms of the left singular vectors $\boldsymbol{U}$ corresponding to the principal components. Inspired by the leverage score in

statistics, we formalize this as Spectral Strength. For a reasoning step $s$ containing multiple tokens, its spectral strength is defined as the average leverage score across all tokens:

$$\mathcal{S}(s) = \frac{1}{|s|} \sum_{t \in s} \|(\boldsymbol{U}_{1:k^*})_t\|_2^2, \qquad (7)$$

where $(\boldsymbol{U}_{1:k^*})_t$ denotes the $t$-th row of the left singular vector matrix. Intuitively, higher spectral strength indicates that a step's gradient aligns more closely with the dominant update directions.

To validate whether this metric captures properties distinct from traditional difficulty metrics, we compare spectral strength against perplexity (PPL). Specifically, we randomly sample 200 instances and compute both metrics for each reasoning step. Since PPL-based filtering typically prioritizes high-PPL steps as informative signals, we focus our analysis on this high-PPL region. We then stratify these steps by spectral strength and qualitatively compare the high spectral strength and low spectral strength subgroups. This allows us to examine whether spectral strength can distinguish semantically meaningful content from noise, even among steps with comparable prediction difficulty.

**Observation.** As illustrated in Figure 3, the results reveal a significant decoupling between spectral strength and perplexity, indicating that our metric captures information orthogonal to prediction difficulty. Notably, we identify a distinct cluster of steps exhibiting ***high PPL but low spectral strength***, which qualitatively corresponds to ambiguous filler phrases or meaningless repetitions. While traditional PPL-based filtering might erroneously prioritize these steps as informative high-difficulty samples, our geometric metric correctly identifies them as non-generalizable noise. This confirms that spectral strength serves as a more faithful indicator of essential reasoning content.

## 2.3. How Do Residual Gradients Affect Model Generalization?

Although we have geometrically isolated the residual subspace, determining its semantic role requires analyzing its statistical properties across samples. We investigate the distribution patterns of these separated residual gradients across different samples to uncover their relationship to model generalization.

**Setting.** To achieve this goal, we analyze the directional consistency of gradients across different samples. Specifically, we randomly sample $1,000$ instances and extract the high spectral strength segments (consensus gradients) and the low spectral strength segments (residual gradients) from each sample. We then compute pairwise cosine similarity between gradients from different samples for both groups, yielding two cross-instance similarity distributions.

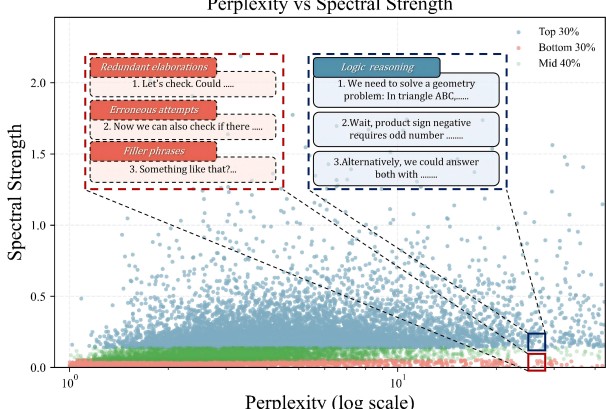

*Figure 3.* **Scatter plot of Spectral Strength ($y$-axis) versus Perplexity ($x$-axis).** The visualization highlights two distinct regions characterized by similarly high perplexity values. The red box identifies steps with low spectral strength (containing redundant elaborations and filler phrases), while the blue box identifies steps with high spectral strength (containing logical reasoning content).

**Observation.** As shown in Figure 4, the results reveal a fundamental difference: Consensus gradients show clear positive correlations across samples, indicating that they capture general reasoning patterns shared across different instances. In contrast, residual gradients exhibit weak, unstructured alignment, with a mean similarity near zero and frequent negative correlations.

These results indicate that residual gradients mainly contain instance-specific information that does not transfer across instances. Their lack of cross-sample consistency causes them to act as conflicting signals during training, cancelling each other out rather than contributing to generalization. This provides empirical evidence and motivation for our subsequent optimization of the training objective to reduce instance-specific interference (Yu et al., 2020; Dandi et al., 2022; Sankararaman et al., 2020).

**Theoretical Perspective.** To rigorously validate these empirical observations, we provide a formal analysis in Appendix D. We model the gradient composition as a signal-plus-noise process, where the reasoning signal resides in a low-rank subspace (Consensus Subspace) and instance-specific interference acts as isotropic noise. We prove that projecting gradients onto the Consensus Subspace mathematically guarantees a reduction in gradient variance by a factor of $k^*/d$ (where $k^*$ is the rank and $d$ is the hidden dimension). This theoretical result confirms that our subspace attribution framework effectively improves the Signal-to-Noise Ratio of the optimization, providing a solid foundation for the subsequent method design.

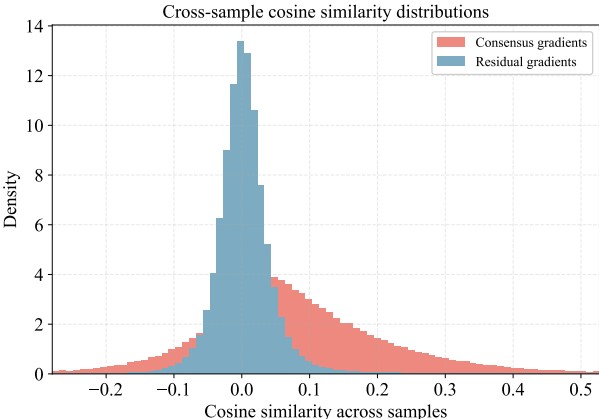

*Figure 4.* **Histograms comparing the cross-sample cosine similarity distributions of consensus and residual gradients.** The $x$-axis represents pairwise cosine similarity, and the $y$-axis represents density. The consensus gradients (red) exhibit a distribution skewed toward positive values, indicating consistent alignment across samples. In contrast, the residual gradients (blue) concentrate symmetrically around zero, exhibiting characteristics of isotropic noise.

## 3. Method

The preceding analysis establishes that high spectral-strength steps drive the consensus subspace, whereas low spectral-strength steps contribute instance-specific residuals that interfere with optimization. Building on this insight, we propose **Spectral-guided Learning**, a selective learning approach that uses spectral strength as the selection criterion to prioritize high-contribution reasoning steps while suppressing instance-specific content.

As illustrated in Figure 5, our selection process consists of the following components:

### 3.1. Gradient Capture and Spectral Analysis

To identify transferable reasoning patterns, we first quantify the contribution of each reasoning step to the consensus subspace.

**Gradient Capture.** We begin by performing standardized step-wise segmentation on the long CoT training data. Using the base model, we capture the gradient representations for each reasoning step. Unlike token-level approaches that break sentence structures, we maintain the granularity of complete reasoning steps to preserve semantic coherence.

**Gradient Spectral Analysis.** With the captured gradients, we perform Singular Value Decomposition (SVD) to construct the consensus subspace. As shown in the middle panel of Figure 5, we decompose the gradient matrix into singular vectors ($U, V^T$) and singular values ($\Sigma$). We analyze the

cumulative energy of the singular values to determine the effective dimensions of the consensus subspace (Truncate at Threshold). Finally, we compute the Spectral Strength for each step by projecting its gradient onto this top-$k$ principal subspace. Steps with high projection magnitude (high spectral strength) are identified as aligning closely with the consensus reasoning patterns.

### 3.2. spectral-guided Selective Learning

Guided by the spectral analysis, our approach selectively trains the model to focus on high-contribution steps, thereby suppressing instance-specific content.

**Dynamic Threshold Truncation.** Since different samples vary significantly in reasoning difficulty and length, adopting a fixed-number selection strategy is not appropriate. We employ a dynamic truncation strategy based on cumulative energy. Within each sample, we rank all reasoning steps by spectral strength in descending order and accumulate their strength values. We select the minimal set of reasoning steps whose cumulative spectral strength reaches a preset energy threshold as the key reasoning path:

$$\mathcal{S}_{\text{sel}} = \arg\min_{S' \subseteq \mathcal{S}} |S'| \quad \text{s.t.} \quad \frac{\sum_{s \in S'} \mathcal{S}(s)}{\sum_{s \in \mathcal{S}} \mathcal{S}(s)} \geq p, \quad (8)$$

where $p$ is the energy threshold. This means we retain only steps with high spectral strength and discard those with low spectral strength that contribute minimally to the consensus subspace.

### 3.3. Masked Training Objective

After determining the crucial reasoning step that constructs the consensus subspace through the dynamic truncation strategy above, we translate this offline selection result into an online training objective.

Unlike standard SFT that uniformly minimizes the next-token prediction loss across all positions, we adopt a selective masked loss function. Specifically, we construct a sequence mask based on the step-selection results and compute the negative log-likelihood loss only for reasoning steps that construct the consensus subspace. For the remaining steps belonging to the residual subspace, their weights in the loss function are set to zero, thus contributing no gradients:

$$\mathcal{L}_{\text{Spectral}}(\theta) = -\frac{1}{Z} \sum_{t=1}^{T} M_t \log P_\theta(y_t \mid y_{<t}, x),$$
$$\text{where} \quad Z = \sum_{t=1}^{T} M_t. \quad (9)$$

$M_t \in \{0, 1\}$ is the precomputed mask indicating whether position $t$ belongs to a selected high spectral strength step.

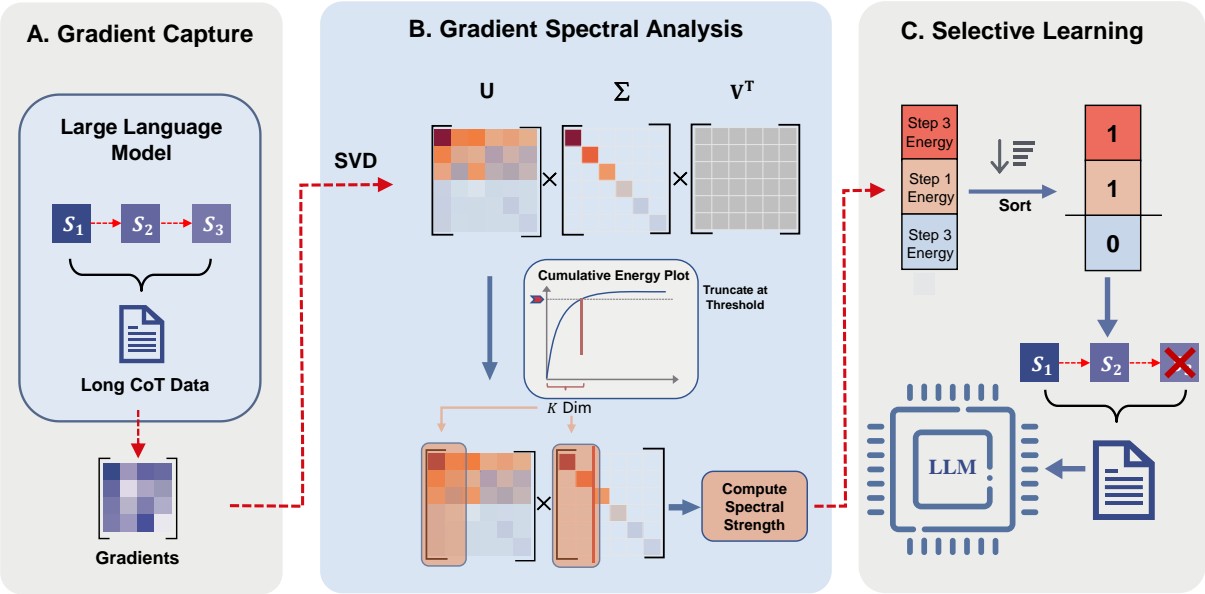

*Figure 5.* **Schematic of the Spectral-guided Learning framework.** The pipeline consists of three stages: (A) Extraction of gradients from a base model using step-wise CoT data. (B) Singular Value Decomposition of gradients to compute spectral strength via projection onto a truncated $k$-dimensional subspace. (C) Selection of reasoning steps based on a cumulative energy threshold, followed by model training using the resulting selective gradient masks.

This selective optimization strategy decouples context modeling from parameter updates. During forward propagation, steps with low spectral strength in the residual subspace still fully participate in the hidden-state computation, ensuring that the model maintains semantic coherence and complete context dependency. During backward propagation, suppressing the gradient contribution of these steps effectively blocks instance-specific residual interference. As a result, parameter updates are guided along the consensus subspace, enabling the model to efficiently learn transferable reasoning patterns while preserving the logical integrity of long chains of thought.

## 4. Experiments

This section validates the effectiveness of Spectral-guided Learning through systematic experiments.

### 4.1. Experimental Setup

**Models.** We evaluate Spectral-guided Learning and comparison methods on Qwen3-4B-Base, Qwen3-8B-Base, Qwen3-4B-Instruct-2507, and Qwen2.5-7B-Instruct (Yang et al., 2025) since they represent diverse parameter scales and alignment types that are widely used in reasoning tasks. All experiments are run on an 8xNVIDIA L20 GPU server.

**Datasets.** To comprehensively evaluate mathematical reasoning ability, we select five representative benchmark

datasets: ***AIME24&25*** (AIME, 2025), the American Invitational Mathematics Examination 2024&2025, which contains 30 high-difficulty competition problems; ***MATH500*** (Hendrycks et al., 2021), which contains 500 problems randomly sampled from the MATH test set, covering seven mathematical domains, used to test fundamental mathematical ability; ***OlympiadBench*** (He et al., 2024), which is a collection of Olympiad-level mathematical problems, used to evaluate reasoning performance in extremely difficult scenarios; ***GPQA-Diamond*** (Rein et al., 2023), which is a graduate-level benchmark of 198 expert-written multiple-choice questions spanning physics, chemistry, and biology, emphasizing "Google-proof" deep academic reasoning.

**Training Data.** For training data construction, we randomly sample 10,000 problems from the AceReason-1.1-SFT dataset (Chen et al., 2025). For these problems, we use DeepSeek-R1-0528 (DeepSeek-AI et al., 2025) to generate high-quality long CoT reasoning trajectories. These data constitute our supervised finetuning dataset.

**Baseline.** We compare Spectral-guided Learning with the following baseline methods: ***Vanilla SFT***: Standard supervised finetuning that trains on all steps in the CoT, serving as the reference upper bound. ***TDST*** (Cui et al., 2025): Target-Distribution Supervision Tailoring(TDST) that tailors supervision to the student model by favoring CoT steps whose distributions align most closely with the student's

*Table 1.* **The accuracy (%) on math and reasoning benchmarks.** Small numbers indicate the performance improvement compared to the Vanilla SFT baseline. **Bold** indicates the best result in this comparison.

| Model | Method | AIME24 | AIME25 | MATH500 | OlympiadBench | GPQA | Avg. |
|---|---|---|---|---|---|---|---|
| Qwen2.5 7B-Instruct | + Vanilla SFT | 24.20 | 21.60 | 79.80 | 44.20 | 33.10 | 40.60 |
| | + EDSP | 15.00 | 16.70 | 80.30 | 44.10 | 31.30 | 37.50 |
| | + TDST | 16.70 | **23.30** | 81.90 | 43.00 | 34.10 | 39.80 |
| | + Spectral-guided Learning | **25.00**$_{\uparrow 0.8}$ | 22.50$_{\uparrow 0.9}$ | **83.50**$_{\uparrow 3.7}$ | **44.80**$_{\uparrow 0.6}$ | **34.60**$_{\uparrow 1.5}$ | **42.10**$_{\uparrow 1.5}$ |
| Qwen3 4B-Base | + Vanilla SFT | 35.00 | 26.70 | 87.00 | 50.70 | 43.80 | 48.60 |
| | + EDSP | 31.70 | 28.30 | 85.90 | 51.30 | 45.50 | 48.50 |
| | + TDST | 37.50 | 26.70 | 86.00 | 50.60 | **47.50** | 49.70 |
| | + Spectral-guided Learning | **39.20**$_{\uparrow 4.2}$ | **30.80**$_{\uparrow 4.1}$ | **87.40**$_{\uparrow 0.4}$ | **51.70**$_{\uparrow 1.0}$ | **47.50**$_{\uparrow 3.7}$ | **51.30**$_{\uparrow 2.7}$ |
| Qwen3 8B-Base | + Vanilla SFT | 45.00 | 29.20 | 88.20 | 54.70 | 52.90 | 54.00 |
| | + EDSP | 44.20 | 31.70 | **89.40** | 54.80 | 54.90 | 55.00 |
| | + TDST | 45.00 | 32.50 | 89.00 | 54.00 | 53.80 | 54.90 |
| | + Spectral-guided Learning | **46.70**$_{\uparrow 1.7}$ | **35.00**$_{\uparrow 5.8}$ | **89.40**$_{\uparrow 1.2}$ | **55.80**$_{\uparrow 1.1}$ | **56.40**$_{\uparrow 3.5}$ | **56.70**$_{\uparrow 2.7}$ |
| Qwen3 4B-Instruct | + Vanilla SFT | 51.70 | 37.50 | 83.90 | 54.40 | 51.10 | 55.70 |
| | + EDSP | 52.50 | 43.30 | **88.60** | 57.60 | 53.20 | 59.00 |
| | + TDST | 49.20 | 43.30 | 85.40 | 57.10 | 53.70 | 57.70 |
| | + Spectral-guided Learning | **59.20**$_{\uparrow 7.5}$ | **45.00**$_{\uparrow 7.5}$ | 88.40$_{\uparrow 4.5}$ | **58.60**$_{\uparrow 4.2}$ | **54.70**$_{\uparrow 3.6}$ | **61.20**$_{\uparrow 5.5}$ |

pretrained distribution. Concretely, it performs a model-fit selection strategy that prioritizes steps with higher length-normalized likelihood (equivalently, lower perplexity) under the student model. ***EDSP*** (Li et al., 2025): Entropy-Driven Supervision Prioritization(EDSP) that prioritizes training signal from CoT steps with higher predictive entropy. Concretely, it performs high-entropy selection, emphasizing high-entropy reasoning steps during training.

**Evaluation Setup.** To ensure the reliability and stability of evaluation results, all evaluations are conducted under unified settings. Specifically, we set the sampling temperature to 0.6 and Top-p to 0.95. For each test problem, we sample 4 responses with a maximum generation length of 32,768 tokens. We report the average accuracy on each benchmark.

### 4.2. Results and Discussion

Table 1 summarizes the performance of various models across four mathematical reasoning benchmarks. Our primary comparison contrasts Vanilla SFT with our Spectral-guided Learning approach. The results indicate that Spectral-guided Learning consistently outperforms Vanilla SFT across most tasks, achieving superior reasoning accuracy while utilizing fewer tokens for parameter updates. For a comprehensive breakdown of experimental results, including comparisons with additional heuristic baselines, please refer to Appendix C.

Spectral-guided Learning yields consistent performance gains across all evaluated architectures. For base models, Qwen3-4B-Base achieves an improvement in average accuracy from 48.6% to 51.3% (+2.7%), while Qwen3-8B-Base improves from 54.0% to 56.7% (+2.7%). The gains are even

more substantial for Instruct models: Qwen3-4B-Instruct-2507 sees a significant boost from 55.7% to 61.2% (+5.5%), and Qwen2.5-7B-Instruct improves from 40.6% to 42.1% (+1.5%). These results demonstrate that our spectral-guided strategy generalizes robustly across different model scales and alignment stages, effectively enhancing reasoning capabilities regardless of the base model's initial competency.

The improvements are particularly pronounced on high-difficulty competition datasets, highlighting the method's ability to distill complex logic. Specifically, on AIME24, Qwen3-4B-Base improves by 4.2% (from 35.0% to 39.2%), while Qwen3-4B-Instruct-2507 achieves a remarkable 7.5% gain (from 51.7% to 59.2%). Similarly, on AIME25, Qwen3-8B-Base demonstrates a 5.8% improvement (from 29.2% to 35.0%). Furthermore, regarding fundamental mathematical proficiency, Qwen3-4B-Instruct-2507 records a notable surge in MATH500 accuracy, rising from 83.9% to 88.4% (+4.5%).

**Discussion.** These empirical results corroborate the theoretical framework presented in Section 2: effective training signals in long CoTs are highly concentrated within a low-rank consensus subspace. By leveraging gradient-based spectral analysis, our method successfully filters out redundant reasoning steps that reside in the residual subspace. Rather than contributing to generalization, these residual segments lack cross-sample consistency and often act as noise that interferes with the optimization landscape.

Crucially, our experiments demonstrate that restricting optimization to high spectral strength steps does not compromise performance; on the contrary, it significantly enhances generalization on complex reasoning tasks. The substan-

*Table 2.* **Ablation study of different selection strategies on Qwen3-4B-Base.** We report accuracy (%) across all five benchmarks. Small numbers indicate the improvement over Vanilla SFT.

| Strategy | AIME24 | AIME25 | MATH500 | OLY. | GPQA | Avg. |
|---|---|---|---|---|---|---|
| Vanilla SFT | 35.0 | 26.7 | 87.0 | 50.7 | 43.8 | 48.6 |
| Random | 33.3 | 25.0 | 86.2 | 49.2 | 41.9 | 47.1 |
| Residuals | 30.8 | 22.5 | 84.5 | 46.8 | 40.2 | 44.9 |
| Static Ratio | 37.5 | 27.8 | 87.1 | 51.0 | 45.4 | 49.8 |
| **Spectral-guided Learning** | $\mathbf{39.2}_{\uparrow 4.2}$ | $\mathbf{30.8}_{\uparrow 2.5}$ | $\mathbf{87.4}_{\uparrow 0.4}$ | $\mathbf{51.7}_{\uparrow 0}$ | $\mathbf{47.5}_{\uparrow 3.}$ | $\mathbf{51.0}_{\uparrow 2.4}$ |

tial gains observed, particularly in Instruct models, suggest that Spectral-guided Learning efficiently unlocks the latent reasoning potential of pre-aligned models, offering a more data-efficient fine-tuning paradigm compared to standard full-parameter training.

### 4.3. Ablation Study

To validate our proposed metric and selection strategy, we conduct ablation studies using Qwen3-4B-Base, summarized in Table 2. For evaluations across additional model scales, see Appendix B.

**Effectiveness of Spectral Strength.** We assess whether spectral strength distinguishes generalizable reasoning from instance-specific noise using two baselines: (1) *Random Selection*: Masking reasoning steps randomly to match our method's average retention ratio. (2) *Residual Selection*: Fine-tuning exclusively on steps with low spectral strength (the residual subspace) that our method typically discards.

As shown in Table 2, random selection decreases performance by 1.5% on average, indicating that token reduction alone does not improve efficiency. Notably, residual selection results in a 3.6% degradation, performing worse than vanilla SFT. This confirms our theoretical insight from Section 2: the residual subspace is dominated by conflicting gradients and noise that impede optimization. In contrast, focusing on the consensus subspace yields substantial gains, proving that spectral strength effectively identifies core reasoning patterns.

**Necessity of Dynamic Thresholding.** Our method uses a dynamic truncation strategy based on cumulative energy to adaptively determine the number of critical steps per sample. We compare this against *Static Selection*, which retains a fixed proportion of top-ranked steps for all instances. While Static Selection outperforms Vanilla SFT, it achieves only a 1.2% average improvement—half the 2.4% gain of our dynamic approach. This performance gap suggests that reasoning complexity is non-uniform; a fixed ratio either truncates necessary logic in complex tasks or retains redundant noise in simpler ones. Our dynamic strategy successfully navigates this trade-off.

## 5. Related Work

**Reasoning traces and efficiency.** Chain-of-Thought prompting and rationale-based distillation effectively enhance multi-step reasoning but introduce significant training costs due to verbose traces (Wei et al., 2022; Zelikman et al., 2022; DeepSeek-AI et al., 2025). To mitigate redundancy, recent works propose pruning steps, controlling token budgets, or internalizing reasoning into latent spaces (Han et al., 2024; Hao et al., 2024; Wu et al., 2025). Our work is orthogonal to these rewriting approaches: instead of compressing traces, we focus on *selecting* steps that provide stable optimization signals, thereby utilizing long traces more effectively.

**Gradient-based selective learning.** Selective optimization methods improve efficiency and robustness by prioritizing data via gradient attribution or loss reweighting (Sundararajan et al., 2017; Hans et al., 2024; Lin et al., 2024). Aligned with findings that effective parameter updates are often low-rank (Hu et al., 2021), we propose a spectral approach distinct from heuristic-based selection. We score reasoning steps by analyzing the spectral structure of their loss gradients, extracting a shared low-rank consensus subspace to suppress noisy updates and enable stable learning from complex traces.

## 6. Conclusion

This paper investigates data efficiency mechanisms in reasoning alignment. We introduce the Loss Subspace Attribution framework to decode gradient geometry, revealing that robust reasoning stems from a low-rank consensus subspace, while residuals are dominated by conflicting noise. Leveraging this, we propose Spectral-guided Learning to distill generalization-critical signals from Chain-of-Thought data. By aligning gradient updates with the consensus, our approach isolates core reasoning from redundancy, outperforming standard fine-tuning and heuristic filtering in accuracy and cost for scalable, interpretable reasoning.

## Acknowledgments

This work was supported in part by the "Pioneer and Leading Goose" R&D Programs of Zhejiang under Grant 2025SSYS0003, and in part by the National Natural Science Foundation of China under Grant 62271452. We also thank Tongyi Lab, Alibaba Group for hosting the internship program during which this research was conducted. The author denoted by † is the project lead.

## Impact Statement

This paper introduces Spectral-guided Learning to improve the efficiency of reasoning distillation for large language models. By identifying a low-rank consensus subspace in gradient geometry, the proposed method isolates core logical reasoning from redundant, instance-specific noise. This selective optimization significantly reduces computational overhead and training costs while consistently boosting the generalization performance of student models across complex mathematical and scientific benchmarks.

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

## A. Training Details

We implement Spectral-guided Learning using the Llama-factory training framework. All models are trained using the configurations detailed in Table 3. We utilize the AdamW optimizer with a cosine learning rate scheduler. To ensure fair comparisons and stable training, we adopt a unified set of hyperparameters across all backbone models. Specifically, all models are trained for 6 epochs with a global batch size of 32 and a maximum sequence length (cutoff length) of 32,768 tokens. We utilize a consistent learning rate of 5e-5 and a warmup ratio of 0.1 across all experiments.

*Table 3.* **Training configuration for different backbone models.**

| Hyperparameter | Qwen3-4B-Base | Qwen3-8B-Base | Qwen3-4B-Instruct-2507 | Qwen2.5-7B-Instruct |
|---|---|---|---|---|
| Learning Rate | 5e-5 | 5e-5 | 5e-5 | 5e-5 |
| Cutoff Length | 32k | 32k | 32k | 32k |
| Epochs | 6 | 6 | 6 | 6 |
| Batch Size | 32 | 32 | 32 | 32 |
| LR Scheduler | cosine_with_min_lr | cosine_with_min_lr | cosine_with_min_lr | cosine_with_min_lr |
| Min LR | 1e-5 | 1e-5 | 1e-5 | 1e-5 |
| Warmup Ratio | 0.1 | 0.1 | 0.1 | 0.1 |

## B. Comprehensive Ablation Studies

To demonstrate the robustness of our method across different model scales and alignment stages, we present a unified ablation study in Table 4. We compare Spectral-guided Learning against Vanilla SFT, Random Selection, and Static Ratio Selection across four different backbone models.

The results consistently show that Spectral-guided Learning achieves the best performance across all architectures. Notably, the gap between static ratio selection and Spectral-guided Learning is significant (e.g., +4.1% on Qwen3-4B-Instruct). These results validate our earlier analysis that, since reasoning difficulty and trajectory length vary significantly across samples, adopting a fixed-ratio selection strategy is suboptimal. By dynamically adjusting the retention threshold based on spectral energy, our method successfully preserves the complete logic chain for complex problems while aggressively pruning redundancy in simpler ones, a flexibility that static methods lack.

*Table 4.* **Unified ablation study across different models.** We compare the average accuracy (%) on five benchmarks.

| Model | Method | AIME24 | AIME25 | MATH | OLY | GPQA | Avg. |
|---|---|---|---|---|---|---|---|
| Qwen2.5-7B-Instruct | Vanilla SFT | 24.2 | 21.6 | 79.8 | 44.2 | 33.1 | 40.6 |
| | Random Selection | 15.0 | 20.0 | 80.9 | 43.3 | 31.6 | 38.2 |
| | Static Ratio Selection | 24.6 | 22.1 | 80.0 | 44.5 | 33.6 | 40.9 |
| | **Spectral-guided Learning** | **25.0** | **22.5** | **83.5** | **44.8** | **34.6** | **42.1** |
| Qwen3-4B-Base | Vanilla SFT | 35.0 | 26.7 | 87.0 | 50.7 | 43.8 | 48.6 |
| | Random Selection | 33.3 | 25.0 | 86.2 | 49.2 | 41.9 | 47.1 |
| | Static Ratio Selection | 37.5 | 27.8 | 87.1 | 51.0 | 45.4 | 49.8 |
| | **Spectral-guided Learning** | **39.2** | **30.8** | **87.4** | **51.7** | **47.5** | **51.3** |
| Qwen3-8B-Base | Vanilla SFT | 45.0 | 29.2 | 88.2 | 54.7 | 52.9 | 54.0 |
| | Random Selection | 41.7 | 26.7 | 88.8 | 51.7 | 51.4 | 52.1 |
| | Static Ratio Selection | 45.8 | 30.5 | 88.5 | 54.8 | 54.1 | 54.7 |
| | **Spectral-guided Learning** | **46.7** | **35.0** | **89.4** | **55.8** | **56.4** | **56.7** |
| Qwen3-4B-Instruct | Vanilla SFT | 51.7 | 37.5 | 83.9 | 54.4 | 51.1 | 55.7 |
| | Random Selection | 50.0 | 35.0 | 81.3 | 54.8 | 47.9 | 53.8 |
| | Static Ratio Selection | 54.5 | 40.2 | 85.0 | 55.8 | 52.8 | 57.7 |
| | **Spectral-guided Learning** | **59.2** | **45.0** | **88.4** | **58.6** | **54.7** | **61.2** |

## C. Additional Experimental Results

In addition to the main baselines, we evaluate two additional selection strategies to further validate the robustness of our criteria: (1) **ESSP** (Entropy-Suppressed Supervision Prioritization): The reverse of EDSP. This method performs low-entropy selection by emphasizing steps with lower predictive entropy during training, effectively prioritizing high-confidence (often simpler) steps. (2) **HTST** (Hard-Target Supervision Tailoring): The reverse of TDST. This method performs hard-fit selection by prioritizing steps with lower length-normalized likelihood (equivalently, higher perplexity) under the student model, tailoring supervision toward harder-to-fit steps. Table 5 presents the detailed performance comparison, including these additional baselines, across all evaluated models.

*Table 5.* **Comprehensive performance comparison across multiple models and baselines.** We report the accuracy (%) on reasoning benchmarks. **Bold** indicates the best result.

| Model | Method | AIME24 | AIME25 | MATH500 | OlympiadBench | GPQA | Avg. |
|---|---|---|---|---|---|---|---|
| Qwen2.5 7B-Instruct | + Vanilla SFT | 24.20 | 21.60 | 79.80 | 44.20 | 33.10 | 40.60 |
| | + Random | 15.00 | 20.00 | 80.90 | 43.30 | 31.60 | 38.20 |
| | + ESSP | 18.30 | 16.70 | **84.50** | 44.40 | 34.10 | 39.60 |
| | + EDSP | 15.00 | 16.70 | 80.30 | 44.10 | 31.30 | 37.50 |
| | + HTST | 23.30 | 23.30 | 80.60 | 43.70 | 33.30 | 40.80 |
| | + TDST | 16.70 | 23.30 | 81.90 | 43.00 | 34.10 | 39.80 |
| | + Spectral-guided Learning | **25.00**$_{\uparrow 0.8}$ | **22.50**$_{\uparrow 0.9}$ | 83.50$_{\uparrow 3.7}$ | **44.80**$_{\uparrow 0.6}$ | **34.60**$_{\uparrow 1.5}$ | **42.10**$_{\uparrow 1.5}$ |
| Qwen3 4B-Base | + Vanilla SFT | 35.00 | 26.70 | 87.00 | 50.70 | 43.80 | 48.60 |
| | + Random | 33.30 | 25.00 | 86.20 | 49.20 | 41.90 | 47.10 |
| | + ESSP | 29.20 | 30.80 | 86.90 | 51.50 | 46.30 | 48.90 |
| | + EDSP | 31.70 | 28.30 | 85.90 | 51.30 | 45.50 | 48.50 |
| | + HTST | 33.30 | **31.70** | 86.10 | 50.20 | 43.80 | 49.00 |
| | + TDST | 37.50 | 26.70 | 86.00 | 50.60 | 47.50 | 49.70 |
| | + Spectral-guided Learning | **39.20**$_{\uparrow 4.2}$ | 30.80$_{\uparrow 4.1}$ | **87.40**$_{\uparrow 0.4}$ | **51.70**$_{\uparrow 1.0}$ | 47.50$_{\uparrow 3.7}$ | **51.30**$_{\uparrow 2.7}$ |
| Qwen3 8B-Base | + Vanilla SFT | 45.00 | 29.20 | 88.20 | 54.70 | 52.90 | 54.00 |
| | + Random | 41.70 | 26.70 | 88.80 | 51.70 | 51.40 | 52.10 |
| | + ESSP | 47.50 | 29.20 | 89.00 | 55.30 | 55.40 | 55.30 |
| | + EDSP | 44.20 | 31.70 | 89.40 | 54.80 | 54.90 | 55.00 |
| | + HTST | 41.70 | 30.80 | 88.50 | 55.30 | 53.70 | 54.00 |
| | + TDST | 45.00 | 32.50 | 89.00 | 54.00 | 53.80 | 54.90 |
| | + Spectral-guided Learning | **46.70**$_{\uparrow 1.7}$ | **35.00**$_{\uparrow 5.8}$ | **89.40**$_{\uparrow 1.2}$ | **55.80**$_{\uparrow 1.1}$ | **56.40**$_{\uparrow 3.5}$ | **56.70**$_{\uparrow 2.7}$ |
| Qwen3 4B-Instruct | + Vanilla SFT | 51.70 | 37.50 | 83.90 | 54.40 | 51.10 | 55.70 |
| | + Random | 50.00 | 35.00 | 81.30 | 54.80 | 47.90 | 53.80 |
| | + ESSP | 58.30 | 42.50 | 75.40 | 58.40 | 46.20 | 56.20 |
| | + EDSP | 52.50 | 43.30 | 88.60 | 57.60 | 53.20 | 59.00 |
| | + HTST | 56.70 | 41.70 | 88.50 | 56.70 | 50.80 | 58.90 |
| | + TDST | 49.20 | 43.30 | 85.40 | 57.10 | 53.70 | 57.70 |
| | + Spectral-guided Learning | **59.20**$_{\uparrow 7.5}$ | **45.00**$_{\uparrow 7.5}$ | **88.40**$_{\uparrow 4.5}$ | **58.60**$_{\uparrow 4.2}$ | 54.70$_{\uparrow 3.6}$ | **61.20**$_{\uparrow 5.5}$ |

## D. Theoretical Analysis of Consensus Gradient Projection

In Section 2, we empirically observed that reasoning signals primarily concentrate in a low-rank subspace, whereas residual gradients exhibit characteristics of isotropic noise. This appendix formalizes this observation within a stochastic optimization framework. We demonstrate that projecting gradients onto the Consensus Subspace functions as a variance reduction mechanism, theoretically guaranteeing an improved Signal-to-Noise Ratio (SNR) for the weight updates.

### D.1. Problem Setup

Let $g_t \in \mathbb{R}^d$ denote the stochastic gradient at step $t$. We model $g_t$ as an unbiased estimator of the true population gradient $\boldsymbol{\mu}$, corrupted by a noise term $\boldsymbol{\xi}$:

$$g_t = \boldsymbol{\mu} + \boldsymbol{\xi}, \quad \text{where } \mathbb{E}[\boldsymbol{\xi}] = \mathbf{0}. \tag{10}$$

Here, $\boldsymbol{\mu}$ represents the ideal update direction driven by generalizable reasoning patterns, while $\boldsymbol{\xi}$ captures instance-specific variations (e.g., lexical artifacts) that do not generalize across samples.

Based on the spectral properties analyzed in Section 2.1, we adopt the following assumptions:

**Assumption 1 (Low-Rank Signal).** The true reasoning gradient $\boldsymbol{\mu}$ lies within the Consensus Subspace $\mathcal{S}_{\parallel}$ of dimension $k^* \ll d$, spanned by the top singular vectors $\boldsymbol{V}_{1:k^*}$. Consequently, the projection operator $\boldsymbol{P}_{\parallel}$ preserves the signal:

$$\boldsymbol{P}_{\parallel}\boldsymbol{\mu} = \boldsymbol{\mu}. \tag{11}$$

**Assumption 2 (Isotropic Noise).** The noise component $\boldsymbol{\xi}$ is isotropic in the high-dimensional parameter space, representing unstructured interference. Its covariance is modeled as:

$$\mathrm{Cov}(\boldsymbol{\xi}) = \mathbb{E}[\boldsymbol{\xi}\boldsymbol{\xi}^{\top}] = \frac{\sigma^2}{d}\boldsymbol{I}_d, \tag{12}$$

where $\sigma^2$ is the total variance of the noise.

### D.2. Variance Reduction Theorem

We define the *Consensus Gradient* as the projection of the raw gradient onto the consensus subspace: $g_t^{\parallel} = \boldsymbol{P}_{\parallel}g_t$.

**Theorem D.1.** *Under Assumptions 1 and 2, the Consensus Gradient $g_t^{\parallel}$ is an unbiased estimator of the true gradient $\boldsymbol{\mu}$, and its total variance is reduced by a factor of $k^*/d$ compared to the raw gradient $g_t$.*

*Proof.* **1. Unbiasedness.** First, we analyze the expectation of the projected gradient:

$$\begin{aligned}\mathbb{E}[g_t^{\parallel}] &= \mathbb{E}[\boldsymbol{P}_{\parallel}(\boldsymbol{\mu} + \boldsymbol{\xi})] \\ &= \boldsymbol{P}_{\parallel}\boldsymbol{\mu} + \boldsymbol{P}_{\parallel}\mathbb{E}[\boldsymbol{\xi}].\end{aligned} \tag{13}$$

By Assumption 1, $\boldsymbol{P}_{\parallel}\boldsymbol{\mu} = \boldsymbol{\mu}$. Since the noise is zero-mean ($\mathbb{E}[\boldsymbol{\xi}] = \boldsymbol{0}$), it follows that:

$$\mathbb{E}[g_t^{\parallel}] = \boldsymbol{\mu} + \boldsymbol{0} = \boldsymbol{\mu}. \tag{14}$$

Thus, the consensus gradient remains an unbiased estimator of the reasoning signal.

**2. Variance Analysis.** The total variance (scalar) of the original stochastic gradient $g_t$ is given by the trace of its covariance matrix:

$$\mathrm{Var}(g_t) \triangleq \mathbb{E}[\|g_t - \boldsymbol{\mu}\|^2] = \mathbb{E}[\|\boldsymbol{\xi}\|^2] = \mathrm{Tr}\left(\frac{\sigma^2}{d}\boldsymbol{I}_d\right) = \sigma^2. \tag{15}$$

Next, we derive the variance of the Consensus Gradient $g_t^{\parallel}$:

$$\mathrm{Var}(g_t^{\parallel}) = \mathbb{E}[\|g_t^{\parallel} - \boldsymbol{\mu}\|^2] = \mathbb{E}[\|\boldsymbol{P}_{\parallel}\boldsymbol{\xi}\|^2]. \tag{16}$$

Using the trace trick $\|x\|^2 = \mathrm{Tr}(xx^{\top})$ and the linearity of expectation:

$$\begin{aligned}\mathrm{Var}(g_t^{\parallel}) &= \mathbb{E}[\mathrm{Tr}(\boldsymbol{P}_{\parallel}\boldsymbol{\xi}\boldsymbol{\xi}^{\top}\boldsymbol{P}_{\parallel}^{\top})] \\ &= \mathrm{Tr}(\boldsymbol{P}_{\parallel}\mathbb{E}[\boldsymbol{\xi}\boldsymbol{\xi}^{\top}]\boldsymbol{P}_{\parallel}) \quad (\text{since } \boldsymbol{P}_{\parallel}^{\top} = \boldsymbol{P}_{\parallel}) \\ &= \mathrm{Tr}\left(\boldsymbol{P}_{\parallel}\left(\frac{\sigma^2}{d}\boldsymbol{I}_d\right)\boldsymbol{P}_{\parallel}\right) \\ &= \frac{\sigma^2}{d}\mathrm{Tr}(\boldsymbol{P}_{\parallel}^2).\end{aligned} \tag{17}$$

Since $\boldsymbol{P}_{\parallel}$ is an orthogonal projection matrix, it is idempotent ($\boldsymbol{P}_{\parallel}^2 = \boldsymbol{P}_{\parallel}$). The trace of a projection matrix equals the dimension of its subspace, i.e., $\mathrm{Tr}(\boldsymbol{P}_{\parallel}) = k^*$. Therefore:

$$\mathrm{Var}(g_t^{\parallel}) = \frac{\sigma^2}{d} \cdot k^*. \tag{18}$$

**Conclusion.** The ratio of variances is:

$$\frac{\text{Var}(g_t^{\parallel})}{\text{Var}(g_t)} = \frac{(k^*/d)\sigma^2}{\sigma^2} = \frac{k^*}{d}. \tag{19}$$

This confirms that the variance reduction is proportional to the dimensionality reduction ratio. $\qquad\square$

**Connection to Empirical Findings.** In our experiments (refer to Figure 2 in Section 2.1), we observed that the gradient energy concentrates heavily in the top singular vectors, implying $k^* \ll d$. Theorem D.1 thus theoretically explains why filtering out the Residual Subspace works: it eliminates the vast majority of the noise variance (specifically, the $\frac{d-k^*}{d}$ fraction) while preserving the core reasoning signal. This aligns perfectly with the superior generalization capability of consensus gradients observed in Section 2.3.

**Remark on Bias-Variance Tradeoff.** We note that Assumption 1 is an idealization. In practice, a negligible fraction of the reasoning signal may spill into the residual subspace. In such cases, $\boldsymbol{P}_{\parallel}$ introduces a small bias. However, given the massive reduction in variance (by orders of magnitude, as $d$ is typically large), the Mean Squared Error (MSE = Bias$^2$ + Variance) is dominated by the variance term. Thus, the proposed projection strictly improves the overall estimation quality of the optimization direction.

