# OpenReview forum: "Uncovering the Gradient Geometry of Long CoT: A Spectral-guided Approach to Reasoning Distillation"
_ICML.cc/2026/Conference — ICML 2026 regular_

### Official Review · Reviewer_RgD1 · 2026-03-09

**Soundness:** 3
**Presentation:** 3
**Significance:** 2
**Originality:** 3
**Overall Recommendation:** 5
**Confidence:** 4

**Summary:**

This paper argues that not all steps in a long chain of thought are worth learning, since many are redundant or instance-specific and do not transfer well across examples.
It introduces Loss Subspace Attribution, which claims that useful reasoning signals concentrate in a low-rank consensus subspace, while the residual subspace is dominated by noisy or conflicting gradients.
Based on this idea, the paper proposes Spectral-guided Learning, which scores reasoning steps by spectral strength and trains mainly on the steps most aligned with the consensus subspace to improve the efficiency and generalization of reasoning distillation.

**Compliance With Llm Reviewing Policy:**

Affirmed.

**Final Justification:**

The rebuttal fully addressed my concerns, and I recommend this paper for acceptance.

**Key Questions For Authors:**

1. Figure 2 clearly illustrates the distinction between reasoning and non-reasoning behaviors.
It would be interesting to know whether the strength of reasoning ability also leads to a systematic trend in the energy curve. In other words, beyond the binary distinction between reasoning and non-reasoning, does the energy trajectory vary in a meaningful way with different levels of reasoning capability?

2. It is unclear whether the negative effect of residual gradients on SFT may diminish as the amount of training data increases.
More concretely, would the proposed method continue to provide consistent gains under data scaling?

**Limitations:**

See Weaknesses and Questions

**Strengths And Weaknesses:**

**Strengths**
1. The paper offers a clear and relatively principled perspective on long-CoT distillation. Instead of relying on surface-level heuristics such as perplexity or entropy, it frames the problem through gradient geometry and argues that transferable reasoning signals concentrate in a low-rank consensus subspace.
2. The paper makes a reasonable methodological choice to operate at the level of complete reasoning steps rather than isolated tokens, and the masked objective is designed to preserve full forward-pass context while suppressing backward interference from low-value steps.

**Weaknesses**
1. **The experimental design could be further strengthened in several respects.**
First, regarding the number of samples, the results on the AIME benchmarks still seem to exhibit some variability with only four samples. It may therefore be helpful to provide additional evidence on the stability of the reported results. Second, if the goal is to more comprehensively demonstrate improvements in reasoning ability, reporting pass@k alongside the current metrics would make the evaluation more complete. Third, the choice of base models appears somewhat narrow, as the experiments focus primarily on the Qwen series. Including Llama-based models could further strengthen the empirical validation, especially given that Llama has also been adopted by DeepSeek as a base model for distillation.

2. **The reported results for Qwen3-4B-Base on the AIME25 dataset appear to differ across Table 1, Table 2, and Table 4.**
It would be helpful if the authors could clarify the reason for these differences.

3. **The paper would also benefit from more detailed analysis of the proposed method.**
For example, it would be informative to report statistics such as the average number of reasoning steps removed per sample. More generally, beyond the single post-training accuracy metric, additional evidence could help better illustrate the substantive benefits of the method. For instance, it would be interesting to know whether the trained model indeed reduces ineffective patterns inherited from the teacher model during inference.

---

> ### Author Rebuttal · Authors · 2026-03-31
>
> ***
> We appreciate your constructive suggestions.
> > **W1:** "First, regarding the number of samples, the results on the AIME benchmarks still seem to exhibit some variability... reporting pass@k alongside the current metrics would make the evaluation more complete. Third, the choice of base models appears somewhat narrow... Including Llama-based models could further strengthen the empirical validation."
>
> **Response:**
> **1. Stability and Pass@k:** To address potential variance from AIME's small size, we evaluated the statistically stable Pass@4 metric. As shown below, our method consistently improves Pass@4 over Vanilla SFT (e.g., Qwen3-4B-Base), confirming our gains are stable. We will add full Pass@k metrics to the appendix.
>
> | Qwen3-4B-Base | AIME24 (Pass@1) | AIME24 (Pass@4) | AIME25 (Pass@1) | AIME25 (Pass@4) |
> | :--- | :--- | :--- | :--- | :--- |
> | Vanilla SFT | 35.0 | 56.7 | 26.7 | 33.3 |
> | **Ours** | **39.2** | **66.7** | **30.8** | **40.0** |
>
> **2. Llama-based Models:** To validate cross-architecture generalization, we evaluated Llama-3-8B-Instruct. Under identical settings, our approach consistently outperforms Vanilla SFT, confirming effective transferability to the Llama architecture.
>
> | Llama-3-8B-Instruct | AIME24 (Pass@1) | AIME24 (Pass@4) | AIME25 (Pass@1) | AIME25 (Pass@4) | MATH500 | OlympiadBench | GPQA-Diamond |
> | :--- | :--- | :--- | :--- | :--- | :--- | :--- | :--- |
> | Vanilla SFT | 10.8 | 20.0 | 3.3 | 13.3 | 62.5 | 30.8 | 30.4 |
> | **Ours** | **12.5** | **16.7** | **5.0** | **13.3** | **64.3** | **31.7** | **32.1** |
> ---
> > **W2:** "The reported results for Qwen3-4B-Base on the AIME25 dataset appear to differ across Table 1, Table 2, and Table 4. It would be helpful if the authors could clarify the reason for these differences."
>
> **Response:**
> We thank the reviewer for catching this typo in Table 2. The correct AIME25 accuracy for Qwen3-4B-Base is **30.8%**, as consistently reported in Tables 1 and 4. The 29.2% was mistakenly transcribed from the baseline. We will correct this.
>
> ---
>
> > **W3:** "For example, it would be informative to report statistics such as the average number of reasoning steps removed per sample... For instance, it would be interesting to know whether the trained model indeed reduces ineffective patterns inherited from the teacher model during inference."
>
> **Response:**
> We will expand our analysis in the revision:
> **1. Removal Statistics:** By explicitly filtering out noise, our method removes an average of 21.2% of reasoning steps per sample, decreasing the average step count from 166 to 131.
> **2. Qualitative Inference:** We will add a Case Study (visuals at: `[https://anonymous.4open.science/api/repo/icml_materials-7303/file/svd_and_sft.png?v=5741779f]`). Vanilla SFT models frequently internalize the teacher's superficial verbosity. In contrast, our method bypasses ineffective patterns, demonstrating a higher density of core logical pivots and more concise reasoning.
>
> ---
>
> > **Q1:** "...does the energy trajectory vary in a meaningful way with different levels of reasoning capability?"
>
> **Response:**
> We investigated this by stratifying CoT data into three quality tiers (High, Medium, Low) based on logical coherence and accuracy. Plotting their SVD cumulative energy curves (figures at: `[https://anonymous.4open.science/api/repo/icml_materials-7303/file/reasoning_capabilities_comparison.png?v=f18039af]`), we observed a systematic trend: trajectories with stronger, rigorous reasoning exhibit a **sharper** curve (higher concentration in top singular values). Conversely, weaker trajectories with excessive trial-and-error exhibit a significantly flatter spectrum. This confirms the consensus subspace's low-rank properties systematically correlate with logical capability.
>
> ---
>
> > **Q2:** "It is unclear whether the negative effect of residual gradients on SFT may diminish as the amount of training data increases. More concretely, would the proposed method continue to provide consistent gains under data scaling?"
>
> **Response:**
> To address whether gains persist under data scaling, we scaled our Qwen3-4B-Base training set from 10k to 20k and 30k instances.
>
> | Qwen3-4B-Base | Method | AIME24 | AIME25 | MATH500 | OlympiadBench | GPQA-Diamond |
> | :--- | :--- | :--- | :--- | :--- | :--- | :--- |
> | **10k** | Vanilla SFT | 35.0 | 26.7 | 87.0 | 50.7 | 43.8 |
> | | **Ours** | **39.2** | **30.8** | **87.4** | **51.7** | **47.5** |
> | **20k** | Vanilla SFT | 36.7 | 29.2 | 87.2 | 51.2 | 41.5 |
> | | **Ours** | **38.3** | **31.7** | **88.3** | **52.1** | **44.2** |
> | **30k** | Vanilla SFT | 40.0 | 30.0 | 87.5 | 52.3 | 42.9 |
> | | **Ours** | **41.7** | **33.3** | **88.5** | **53.2** | **45.1** |
>
> As shown, our method continues to provide significant gains over Vanilla SFT at all scales. This empirically demonstrates that the negative effect of residual gradients is not simply averaged out by larger data volumes; our optimization efficiency scales consistently alongside data growth.

---

> > ### Author Rebuttal · Reviewer_RgD1 · 2026-04-02
> >
> > Thank you to the authors for the clarification and additional experiments. I will increase my score to 5.

---

> > > ### Author Response · Authors · 2026-04-03
> > >
> > > Thanks once again for your valuable comments. We are delighted to hear that your concerns have been resolved, and we are very grateful for your consistently positive evaluation of our work.

---

### Official Review · Reviewer_ainv · 2026-03-12

**Soundness:** 4
**Presentation:** 2
**Significance:** 3
**Originality:** 3
**Overall Recommendation:** 5
**Confidence:** 4

**Summary:**

This paper introduces Spectral-guided Learning to enhance transferable CoT reasoning in small models. By decomposing the gradient space into a low-rank consensus subspace (high-spectral-strength signals) and a high-rank residual subspace (sample-specific noise), the authors identify that uniform SFT leads to noise interference. They propose a dynamic truncation strategy that uses spectral strength to select key reasoning steps; while the forward pass maintains full context, the backward pass updates only these critical gradients, effectively suppressing noise and boosting generalization.

**Compliance With Llm Reviewing Policy:**

Affirmed.

**Final Justification:**

Thanks the authors for the clarification and additional experiments. Thus, I will increase my score to 5.

**Key Questions For Authors:**

1. Could the authors provide a detailed comparison of the training time for each method? Additionally, are there significant differences in the average output length and inference latency of the models trained via Spectral-guided Learning compared to the baselines?
2. What are the specific characteristics of the responses generated by models trained with Spectral-guided Learning? Specifically, is there evidence to show a higher density of "effective reasoning steps" in the final output compared to models trained with standard SFT?
3. If the authors were to perform Loss Subspace Attribution on the outputs of models trained with different methods, would there be a measurable shift in spectral strength? Does Spectral-guided Learning successfully lead to a model whose own gradients are more concentrated within the consensus subspace?

**Limitations:**

The scope of the experiments and discussions is limited to mathematical and scientific tasks. The paper does not explore other reasoning-intensive domains, such as code generation or complex logical reasoning tasks, leaving the cross-domain generalizability of the proposed method unverified.

**Strengths And Weaknesses:**

## Strengths
1. **Theoretical Rigor**: The mathematical derivation is comprehensive and rigorous. The transition from Loss Subspace Attribution theory to the proposed Spectral-guided Learning method is logically sound and well-justified.
2. **Innovative Methodology**: Spectral-guided Learning is highly innovative. The authors clearly articulate the limitations of existing methods and provide a robust comparative analysis through extensive experiments.
3. **Strong Empirical Support**: The ablation studies demonstrate the effectiveness of the proposed step selection. Specifically, Spectral-guided Learning achieves the best performance, while "Residual Selection" (fine-tuning exclusively on low-spectral-strength steps) yields the poorest results, reinforcing the rationality of focusing on the consensus subspace.

## Weaknesses
1.  **Incomplete Performance Metrics**: The experimental results are somewhat brief. The paper lacks a detailed comparison of training time across different methods, as well as the inference-side metrics (e.g., output sequence length and inference latency) for the resulting models.
2.  **Lack of Qualitative Analysis**: There is no case study comparing the outputs of models trained with different methods. This makes it difficult to intuitively understand how the specific reasoning steps selected by Spectral-guided Learning differ from others in practice.
3.  **Formatting Issues in Tables**: Several tables (Table 1, Table 2, and Table 5) have formatting problems where some of the red-colored numerical values are partially cut off or not fully displayed.

---

> ### Author Rebuttal · Authors · 2026-03-31
>
> We sincerely appreciate your time and effort in reviewing our paper and are glad to provide detailed responses to your insightful questions and suggestions.
> >**Q1 & W1:** "Could the authors provide a detailed comparison of the training time for each method? Additionally, are there significant differences in the average output length and inference latency...?"
>
> **Response to Q1 & W1:** We thank the reviewer for suggesting these practical metrics. We will add a comprehensive efficiency analysis to the revised manuscript.
>
> **1. Training Time:** Our method introduces zero additional overhead during the actual SFT phase. The SVD computation is performed entirely offline prior to training. During optimization, the masked loss objective simply zeros out gradients for specific tokens. Empirically, training Qwen3-4B-base on our dataset took approximately 20.67 hours for Vanilla SFT and **18.81** hours for our approach .
>
> **2. Output Length and Inference Latency:** Interestingly, models trained with this spectral-guided objective exhibit a shorter average output length and, consequently, lower inference latency. By penalizing redundant and instance-specific noise during training, the resulting model learns to reason more concisely. On MATH500, the average output length for the Vanilla SFT model was **4419** tokens, whereas ours averaged **4039** tokens, directly leading to a **8.60%** reduction in inference latency.
>
>
> > **Q2 & W2:** "What are the specific characteristics of the responses generated by models trained with Spectral-guided Learning? Specifically, is there evidence to show a higher density of 'effective reasoning steps'..."
>
> **Response to Q2 & W2:** We thank the reviewer for raising this question. To provide evidence of these characteristics, we will add a qualitative Case Study section to the appendix of the revised manuscript, providing side-by-side comparisons of model outputs (which can be viewed via this anonymous link: `[https://anonymous.4open.science/api/repo/icml_materials-7303/file/svd_and_sft.png?v=5741779f]`).
>
> The most prominent characteristic of models trained with our method is indeed a visibly higher density of "effective reasoning steps." While Vanilla SFT models often mimic the superficial verbosity of the teacher, generating repetitive filler phrases (e.g., "Let me think about this again..." or over-explaining simple arithmetic), our model skips these redundant transitions and moves directly between core logical pivots. Because the proposed method masks out the gradients of low-spectral-strength steps during training, the resulting model avoids internalizing these non-essential generation habits.
>
>
>
> > **W3:** "Formatting Issues in Tables: Several tables (Table 1, Table 2, and Table 5) have formatting problems where some of the red-colored numerical values are partially cut off..."
>
> **Response to W3:** We thank the reviewer for pointing out this formatting issue. The red-colored numerical values, which represent the performance improvements over the baseline, inadvertently exceeded the column margins during LaTeX compilation. We have adjusted the column widths and font sizes to ensure all tables are fully readable in the revised manuscript.
>
>
> > **Q3:** "If the authors were to perform Loss Subspace Attribution on the outputs of models trained with different methods, would there be a measurable shift in spectral strength? Does Spectral-guided Learning successfully lead to a model whose own gradients are more concentrated...?"
>
> **Response to Q3:**
>
>
> We thank the reviewer for this insightful question. Prompted by your suggestion, we conducted a post-hoc Loss Subspace Attribution on a subset of data using both the Vanilla SFT baseline and the model trained with our method.(which can be viewed via this anonymous link: `[https://anonymous.4open.science/api/repo/icml_materials-7303/file/method_comparison.png?v=a6533130]`).
>
> We observed a measurable shift in the gradient spectrum: the cumulative energy curve of our model is demonstrably steeper and reaches the 90% energy threshold at a significantly lower rank k compared to the Vanilla SFT model.
>
> This empirical evidence indicates that our method successfully regularizes the model's internal representations. By strictly aligning weight updates with the consensus subspace during training, the resulting model intrinsically learns a low-rank, highly structured hypothesis space for reasoning, inherently producing more concentrated gradients for future inputs. We will add this finding and the corresponding visualization to the Discussion section of the revised manuscript.

---

> > ### Author Rebuttal · Reviewer_ainv · 2026-04-03
> >
> > Thanks the authors for the clarification and additional experiments. I will increase my score to 5.

---

> > > ### Author Response · Authors · 2026-04-03
> > >
> > > Thanks once again for your valuable comments. We are delighted to hear that your concerns have been resolved, and we are very grateful for your consistently positive evaluation of our work.

---

### Official Review · Reviewer_jEYb · 2026-03-18

**Soundness:** 3
**Presentation:** 3
**Significance:** 2
**Originality:** 4
**Overall Recommendation:** 5
**Confidence:** 4

**Summary:**

This paper aims to improve the efficiency of long CoT distillation by identifying reasoning steps that are less informative in building the final output.

The authors perform a gradient decomposition analysis on CoTs, and observe that some steps are more informative than others, in the sense that the gradients corresponding to effective reasoning predominantly lie within a low-rank "consensus subspace", while the residual subspace consists of noisy or conflicting signals.

The training objective is then updated with a mask that only accounts for highly informative reasoning steps (with a "spectral strength" higher than a threshold).

The paper reports consistent gains over vanilla SFT and several heuristic step-selection baselines on multiple student models and reasoning benchmarks.

**Compliance With Llm Reviewing Policy:**

Affirmed.

**Final Justification:**

The rebuttal addressed my main concerns, adding further experiments to strengthen the results.

**Key Questions For Authors:**

1. The main control compares full-CoT with "final answer only" supervision, which confounds reasoning content with sequence length. Could you include a control with long non-reasoning text to better isolate the effect of reasoning content vs. sequence length?
2. Several benchmarks are very small (e.g. AIME with 30 problems x 4 samples), and reported gains are small enough that noise could explain part of them. Can you address this either with larger benchmarks or more robust methods? Possibly, can you include different domains and add more baselines as well?
3. Can you provide more implementation details, e.g. how reasoning steps are segmented and how the threshold for spectral strength is chosen?

**Limitations:**

The authors do not discuss limitations, some limitations include those mentioned in the section above. The impact statement is also missing.

**Strengths And Weaknesses:**

Soundness:

- The motivation is clear and well-articulated: not all tokens in a CoT are equally informative, and treating them uniformly can lead to suboptimal learning.
- The gradient decomposition analysis is a novel and coherent way to identify which steps are more informative for learning, as well as the proposed method of masking the loss based on spectral strength.
- The empirical evaluation covers multiple student models and reasoning benchmarks against several baselines.
- The main control compares full-CoT with "final answer only" supervision, which confounds reasoning content with sequence length. It could be useful to control for that with long non-reasoning text.
- Several benchmarks are very small (e.g. AIME with 30 problems x 4 samples), and reported gains are small enough that noise could matter in the results.

Presentation:

- The paper is very easy to follow, with a coherent narrative and flow.
- Some key implementation details are missing, e.g. how reasoning steps are segmented, or how the threshold is chosen.

Significance:

- Improving data efficiency and generalization of LRM distillation is a highly relevant problem for LM distillation.
- Experiments are limited to mathematical reasoning with a single teacher (DeepSeek-R1-0528). Generalizability to other domains (code, common sense) and teachers is unclear.
- Some baselines are missing, e.g. token-level reweighting methods, CoT compression/summarization, RL-based. The authors claim these methods are orthogonal, but this claim should be better justified.

Originality:

- The idea of using gradient spectral analysis to filter CoT steps is highly novel.

---

> ### Author Rebuttal · Authors · 2026-03-31
>
> We sincerely appreciate your careful review and constructive suggestions.
>
> > **Q1 & W1:** "The main control compares full-CoT with 'final answer only' supervision, which confounds reasoning content with sequence length. Could you include a control with long non-reasoning text to better isolate the effect of reasoning content vs. sequence length?"
>
> **Response:**
> To isolate reasoning content from sequence length, we conducted the suggested control experiment (detailed visualizations at `[https://anonymous.4open.science/api/repo/icml_materials-7303/file/long%20text.png?v=45f96511]`). We sampled 1,000 long factual/descriptive texts matching our CoT dataset's average length and computed their gradient SVD. The gradient spectrum of these non-reasoning texts is significantly flatter, closely resembling the "Non-Reasoning" curve in Figure 2. This robustly confirms the low-rank "consensus subspace" emerges intrinsically from structured reasoning logic, not merely from sequence length.
>
> ---
>
> > **Q2 (Part 1):** "Several benchmarks are very small (e.g. AIME)... reported gains are small enough that noise could explain part of them. Can you address this either with larger benchmarks... Possibly, can you include different domains...?"
>
> **Response:**
> Our evaluation deliberately includes larger-scale and cross-domain benchmarks to rule out statistical noise:
>
> 1. **Stable Metrics (Pass@4):** To mitigate AIME's high Pass@1 variance, we evaluated Pass@4. Our method improves AIME24 Pass@4 from 56.7% to **66.7%** and AIME25 from 33.3% to **40.0%** (Qwen3-4B-Base), proving substantial and stable gains.
> 2. **Large-Scale & Cross-Domain:** We achieve consistent gains on larger datasets like MATH500 (+4.5% on Qwen3-4B-Instruct). Furthermore, **GPQA-Diamond** is a strictly non-mathematical, graduate-level science benchmark. Our gains here (+1.5% to +3.6%) empirically prove generalizability to complex domains beyond pure math.
>
> ---
>
> > **Q2 (Part 2) & W5:** "...add more baselines as well? Some baselines are missing, e.g. token-level reweighting methods, CoT compression/summarization... The authors claim these methods are orthogonal, but this claim should be better justified."
>
> **Response:**
> We evaluated two additional baselines: **Gradient Norm** (selects steps with the highest $L_2$ gradient norm) and **CoT Compression [1]** (physically removes less informative tokens).
>
> | Qwen3-4B-Base   | AIME24   | AIME25   | MATH500  | OlympiadBench | GPQA-Diamond |
> | :-------------- | :------- | :------- | :------- | :------------ | :----------- |
> | Vanilla SFT     | 35.0     | 26.7     | 87.0     | 50.7          | 43.8         |
> | Gradient Norm   | 34.2     | 29.2     | 86.9     | 48.9          | 46.8         |
> | CoT Compression | 31.7     | 25.8     | 86.4     | 51.2          | 44.9         |
> | **Ours**        | **39.2** | **30.8** | **87.4** | **51.7**      | **47.5**     |
>
> Our method consistently outperforms them. Regarding orthogonality: CoT compression operates at the **data level** (shortening inputs). Our method operates at the **optimization level**: we maintain full context during the forward pass (preserving semantic coherence) but selectively mask gradients during the backward pass. Because they target different levels, they can be seamlessly combined. We will explicitly clarify this in Section 5. *(Ref: [1] Making Slow Thinking Faster: Compressing LLM Chain-of-Thought via Step Entropy)*
>
> ---
>
> > **Q3 & W3:** "Can you provide more implementation details, e.g. how reasoning steps are segmented and how the threshold for spectral strength is chosen?"
>
> **Response:**
> We have added these operational details to the revision:
>
> * **Segmentation:** We split CoT trajectories at logical boundaries using regex `r'([.?!\}\]])([\s\n]+)([A-Z])'`, deterministically splitting text after sentence-ending punctuation to preserve semantic completeness.
> * **Threshold Selection:** We set the cumulative energy threshold p = 0.95 across all datasets (Eq. 8). This ensures selected steps capture 95% of the consensus subspace's gradient energy, optimally balancing noise filtration and logic preservation.
>
> ---
>
> > **W4:** "Experiments are limited to mathematical reasoning with a single teacher (DeepSeek-R1-0528). Generalizability to other teachers is unclear."
>
> **Response:**
> To prove our method is not overfitted to a specific teacher's style, we evaluated **gpt-oss-120B** under identical settings:
>
> | Teacher: gpt-oss-120B | AIME24   | AIME25   | MATH500  | OlympiadBench | GPQA-Diamond |
> | :-------------------- | :------- | :------- | :------- | :------------ | :----------- |
> | Vanilla SFT           | 32.5     | 29.2     | 77.8     | 46.2          | 43.1         |
> | **Ours**              | **38.2** | **30.8** | **85.2** | **49.3**      | **47.2**     |
>
> Our method consistently outperforms Vanilla SFT, confirming our spectral metrics are robust across different teacher generation styles.

---

> > ### Author Rebuttal · Reviewer_jEYb · 2026-04-03
> >
> > Thank you for the detailed and thorough responses.
> >
> > Q1: thank you for adding the control experiment, this is very valuable. This concern is resolved.
> >
> > Q2 Part 1: the Pass@4 results on AIME are helpful. However, I note that the AIME benchmarks remain inherently small (30 problems), and Pass@4 still involves limited samples, even though the evidence is stronger now. The gains on MATH500 and GPQA-Diamond are more convincing given their larger size. I consider this concern mostly resolved.
> >
> > Q2 Part 2: the gradient norm and CoT compression baselines are valuable as well. The clarification about orthogonality is convincing and should indeed be made explicit in the paper. This concern is resolved.
> >
> > Q3: thank you for adding details. It could be interesting to see a sensitivity analysis on p, but the concern is largely resolved.
> >
> > W4: thank you for the additional experiment. The concern is resolved.
> >
> > My concerns are mostly resolved, I will increase my score to 5.

---

> > > ### Author Response · Authors · 2026-04-04
> > >
> > > Thanks once again for your valuable comments. We are delighted to hear that your concerns have been resolved, and we are very grateful for your consistently positive evaluation of our work.

---

### Official Review · Reviewer_R4qo · 2026-03-23

**Soundness:** 2
**Presentation:** 2
**Significance:** 2
**Originality:** 3
**Overall Recommendation:** 4
**Confidence:** 4

**Summary:**

This paper tackles how we fine-tune models using long chain-of-thought (CoT) data. The authors argue that we shouldn't treat every part of a long reasoning chain equally because th useful learning signal is highly concentrated within a narrow low-rank gradient subspace, called "consensus subspace". They show that CoT gradients are much more spectrally concentrated than standard answer-only gradients. To take advantage of this, they introduce a method called Spectral-guided Learning. Instead of training on the entire CoT sequence uniformly, they filter it based on quality. They chop the CoT into individual reasoning steps. They project the gradients onto the consensus subspace to measure their spectral strength. They rank the steps by this spectral score and only keep the strongest ones needed to cross a specific threshold. During SFT, the weaker reasoning steps are masked out from the next-token loss calculation. The authors tested this approach on Qwen models using math benchmarks, and the results are promising. By focusing only on the spectrally strong steps, the method outperforms standard SFT as well as a variety of other filtering baselines, like picking steps randomly, looking only at residuals, or just truncating at a static ratio.

**Compliance With Llm Reviewing Policy:**

Affirmed.

**Key Questions For Authors:**

**1. What is the exact operational definition of a “reasoning step”?**

**2. How exactly is the consensus subspace computed during training?**
It is unclear whether the SVD is computed once offline, recomputed during training, or estimated on subsets/batches, and whether it is based only on the student’s initialization or updated as the model changes. A more precise description of when and over what data this subspace is computed would help.

**3. Can the authors support the stronger claim made around Figure 3 more directly?**
Figure 3 shows that spectral strength is not redundant with perplexity, but it does not justify the stronger claim that low-spectral-strength/high-PPL steps are “non-generalizable noise” or that spectral strength faithfully identifies essential reasoning. Could the authors soften this claim or add a matched-budget comparison against random/PPL/entropy selection?

**Limitations:**

No. The paper does not meaningfully discuss its main methodological limitations. A better limitations section should acknowledge that the method may overfit to teacher formatting/style, could amplify biases in teacher-generated CoT traces by selectively reinforcing recurring patterns.

**Strengths And Weaknesses:**

## Strengths

* **Tackling a real, practical issue:** the authors focus on a very relevant problem in reasoning distillation. Standard SFT treats all reasoning tokens equally, even though long chain-of-thought traces are usually full of redundancy, filler, and corrected detours. Framing this as a need for *selective* supervision makes perfect sense for how we actually train reasoning models today.

* **A genuinely fresh perspective:** I like that they moved away from basic perplexity or entropy-style filtering. Looking at CoT distillation through the lens of gradient geometry is a novel, interesting angle. Their Loss Subspace Attribution idea gives the whole paper a a conceptual hook.

* **Simple and actionable:** the method is very simple and practical: capture the step-level gradients, run SVD, score the steps by spectral strength, and then just train with a masked loss on the best ones.




## Weaknesses

* **The paper’s central interpretation is stronger than the evidence shown.**
The key claim is not the SVD itself, but the interpretation of the top singular subspace as a “consensus subspace” containing transferable reasoning, with the orthogonal complement treated as residual noise. However, Section 2.1 only shows that pooled CoT gradients have a more concentrated spectrum than the non-reasoning baseline, i.e. that a dominant low-rank subspace exists.   Section 2.3 further shows that high-spectral-strength gradients are more aligned across samples than low-spectral-strength ones, which is suggestive but still compatible with simpler explanations such as repeated teacher style or formatting regularities.  Since a top principal subspace always exists by construction, the real issue is its interpretation, and that step feels assumed more than demonstrated. I think this is my main concern with this paper.


* **Important gradient-based baselines are missing.**
The baseline section compares primarily against Vanilla SFT, TDST and EDSP.  The issue is that these baselines do not really test the paper’s main claim, which is about the value of a consensus-subspace view of gradients. In Section 2, the paper argues that useful reasoning lies in a shared low-rank gradient subspace and then builds the method around spectral strength derived from that subspace.   Given that framing, the natural missing comparisons are to simpler gradient-based selectors, such as step-wise gradient norm, mean cross-sample gradient similarity, or a gradient-spectrum/diversity criterion such as G-Vendi, which measures diversity via the entropy of gradient covariance [1]. Without such baselines, it remains unclear whether the gains are specific to the proposed consensus-subspace construction or would also arise from a more generic gradient-aware filtering rule. I suggest to the authors to add one simple gradient-based baseline under the same retention ratio; even a per-step gradient-norm selector would materially strengthen the comparison.


* **The notion of a “reasoning step” is under-specified.**
Section 3 says the method begins with “standardized step-wise segmentation” and operates on “complete reasoning steps,” but it never clearly defines what counts as a step or how step boundaries are obtained in practice.   Without a precise segmentation rule, an important source of methodological variability remains unresolved. I suggest that the authors explicitly specify the segmentation procedure and add a small sensitivity check with an alternative segmentation heuristic.

* **Figure 3 shows disagreement with PPL, but not “non-generalizable noise.”**
Figure 3 analysis is qualitative: the paper samples 200 instances, focuses on the high-PPL region, and visually compares high- vs low-spectral-strength steps.  This is enough to show that spectral strength is not redundant with perplexity, but not enough to conclude that low-spectral-strength/high-PPL steps are “non-generalizable noise” or that spectral strength is a faithful measure of essential reasoning content. No direct step-level measure of transferability or causal contribution is provided. This is another overclaim made by the authors and i recommend  them to soften it.



[1] Prismatic Synthesis: Gradient-based Data Diversification Boosts Generalization in LLM Reasoning

---

> ### Author Rebuttal · Authors · 2026-03-31
>
> We appreciate your insightful feedback.
>
> > **W1:The key claim is not the SVD itself, but the interpretation of the top singular subspace as a 'consensus subspace'... compatible with simpler explanations such as repeated teacher style or formatting regularities.**
>
> **Response:**
> We agree and clarify our core interpretation.
>
> Our logic rests on a geometric discovery: long CoT gradients exhibit a prominent low-rank structure. Unlike Perplexity, spectral strength captures intrinsic optimization characteristics. Our theoretical proof (Appendix D) confirms projecting gradients onto the consensus subspace guarantees variance reduction, making these steps strictly beneficial for optimization.
>
> To test the "formatting regularities" hypothesis, we first used a strong LLM (Minimax-2.7) to inductively summarize 1,000 sampled steps, deriving common reasoning and noise categories. We then tasked the model to blindly classify steps across subspaces into these derived categories:
>
> **Table 1: Distribution of Reasoning Patterns Across Subspaces**
>
> | **Subspace** | **Core Reasoning** | Logical Ded. | Math Trans. | Prob. Decomp. | Hypo. Test | **Redundancy/Noise** | Filler | Superficial | Repeat Calc. | Truncated |
> | :--- | :--- | :--- | :--- | :--- | :--- | :--- | :--- | :--- | :--- | :--- |
> | **Consensus** | **81.6%** | 31.2% | 24.5% | 15.1% | 10.8% | **18.4%** | 6.5% | 5.2% | 4.1% | 2.6% |
> | **Residual** | **28.4%** | 8.5% | 10.2% | 6.1% | 3.6% | **71.6%** | 30.5% | 19.4% | 13.2% | 8.5% |
>
> The consensus subspace focuses on core reasoning (81.6%), while the residual is noise-dominated (71.6%). To directly address the "teacher style" concern, a further binary classification on consensus steps confirms this:
>
> **Table 2: Classification of Consensus Subspace Steps**
>
> | **Analyzed Object** | **Yes (Formatting / Teacher Style)** | **No** |
> | :--- | :--- | :--- |
> | **Consensus Subspace** | **5.8%** | **94.2%** |
>
> Only 5.8% of the consensus subspace constitutes stylistic artifacts, proving it captures transferable logic. This is corroborated by major gains on Out-Of-Distribution (OOD) tasks requiring deep logic, like AIME24 (+7.5% for Qwen3-4B-Instruct) and GPQA-Diamond (+3.6%). We will discuss stylistic overfitting risks in our "Limitations".
>
> ---
>
> > **W2:** *"Add one simple gradient-based baseline under the same retention ratio."*
>
> **Response:**
> We evaluated two baselines using our exact dynamic retention ratio:
> 1. **Gradient Norm:** Selects steps with the highest gradient $L_2$ norm ($||g_t||_2$).
> 2. **CoT Compression [1]:** Physically removes less informative steps.
>
> **Table 3: Comparison with Gradient Norm Baseline (Qwen3-4B-Base)**
>
> | Method | AIME24 | AIME25 | MATH500 | OlympiadBench | GPQA-Diamond |
> | :--- | :--- | :--- | :--- | :--- | :--- |
> | Vanilla SFT | 35.0 | 26.7 | 87.0 | 50.7 | 43.8 |
> | Gradient Norm | 34.2 | 29.2 | 86.9 | 48.9 | 46.8 |
> | CoT Compression | 31.7 | 25.8 | 86.4 | 51.2 | 44.9 |
> | **Ours** | **39.2** | **30.8** | **87.4** | **51.7** | **47.5** |
>
> While Gradient Norm marginally improves Vanilla SFT, it consistently underperforms our approach. This confirms gradient magnitude alone cannot guarantee generalizability; the consensus subspace's structured alignment is crucial.
> *(Ref: [1] Making Slow Thinking Faster: Compressing LLM Chain-of-Thought via Step Entropy)*
>
> ---
>
> > **W3/Q1/Q2:** *"What is the exact operational definition of a 'reasoning step'? How exactly is the consensus subspace computed during training?"*
>
> **Response:**
> We have added these details to the revision:
> * **Reasoning Step Definition:** We split CoT trajectories at logical boundaries using regex `r'([.?!\}\]])([\s\n]+)([A-Z])'`, deterministically splitting text after sentence-ending punctuation to preserve semantic completeness.
> * **Consensus Subspace Computation:** SVD is computed **offline, exactly once** before training, using the student's initial weights. Masks are statically applied during SFT with zero computational overhead.
>
> ---
>
> > **W4/Q3:** *"Figure 3 shows disagreement with PPL, but not 'non-generalizable noise.' Could the authors soften this claim or add a matched-budget comparison?"*
>
> **Response:**
> We acknowledge 'non-generalizable noise' was overly definitive and softened it to *"steps with low alignment to the dominant optimization direction."* However, we support our causal claim with two pieces of evidence:
> 1. **Semantic Evidence:** Table 1 proves the residual subspace is dominated by redundant/noisy patterns (71.6%).
> 2. **Causal Evidence:** In our matched-budget Ablation Study, training exclusively on low-spectral-strength steps caused noticeable degradation vs. Vanilla SFT (e.g., dropping from 48.6% to 44.9% on Qwen3-4B-Base). This causally proves updating parameters on residual steps impedes optimization.

---

> > ### Author Rebuttal · Reviewer_R4qo · 2026-03-31
> >
> > the authors added the baselines i was caring about (the gradient baseline) and replied to my questions. I will increase my score by 1 point from 3 to 4.

---

> > > ### Author Response · Authors · 2026-04-01
> > >
> > > Thanks once again for your valuable comments. We are delighted to hear that your concerns have been resolved, and we are very grateful for your positive evaluation of our work.

---

### Decision · Program_Chairs · 2026-04-30

**Decision:**

Accept (regular)

**Comment:**

The paper posits that standard SFT on long-CoT is sub-optimal since it weighs all tokens equally. This is tested through a gradient analysis method which shows that gradients from useful reasoning chains lie in a low-rank subspace, while outside of this subspace, it is gradients from the remaining noisier tokens that dominate. The paper proposes a method to leverage this insight to train better student models.

All reviewers appreciated the approach here, of looking at distillation through gradient geometry, as highly innovative, simple, clear and principled. Multiple concerns were raised in the initial reviews, which the authors seem to have convincingly addressed. This includes some key baselines (`Rq4o`, `jEYb`), missing metrics (`ainv`), ablations with non-reasoning text (`jEYb`), generalization to other domains (`jEYb`), missing experimental details (`jEYb`) and other qualitative details (`RgD1`). The authors' response has been comprehensive on this regard (and all reviewers have acknowledge and updated their scores accordingly). Overall, we are happy to recommend acceptance of this work in particular for its originality and simplicity of its idea.